# Study on Short-Circuiting GMAW Pool Behavior and Microstructure of the Weld with Different Waveform Control Methods

**Tao Chen [1], Songbai Xue [1,\*] , Bo Wang [2], Peizhuo Zhai [1] and Weimin Long [2]**

1 College of Materials Science and Technology, Nanjing University of Aeronautics and Astronautics, Nanjing 210016, China; taocmsc@nuaa.edu.cn (T.C.); zhaipz@nuaa.edu.cn (P.Z.)
2 China Intelligent Equipment Innovation Institute (Ningbo) Co., Ltd., Ningbo 315700, China; wangbo4175@126.com (B.W.); brazelong@163.com (W.L.)
\* Correspondence: xuesb@nuaa.edu.cn; Tel.: +86-8489-6070

**Abstract:** In order to study internal relation among the behavior of the weld pool, the microstructure of weld bead and the waveform of short-circuiting gas metal arc welding (S-GMAW), a high speed photograph-images analysis system was formed to extract characteristics of weld pool behavior. Three representative waveform control methods were used to provide partly and fully penetrated weld pools and beads. It was found that the behavior of the weld pool was related to the instantaneous power density of the liquid bridge at the break-up time. Weld pool oscillation was triggered by the explosion of the liquid bridge, the natural oscillation frequencies were derived by the continuous wavelet transform. The change of weld pool state caused the transition of oscillation mode, and it led to different nature oscillation frequencies between partial and full penetration. Slags flow pattern could be an indication of the weld pool flow. Compared with the scattered slags on fully penetrated weld pool, slag particles accumulated on partially penetrated weld pools. The oscillating promoted the convection of the welding pool and resulted in larger melting width and depth, the grain size, and the content of pro-eutectoid ferrite in the weld microstructure of S235JR increased, the content of acicular ferrite decreased.

**Keywords:** short-circuiting gas metal arc welding; waveform control method; weld pool oscillation and flow; microstructure; high speed photograph; image processing; continuous wavelet transform

---

## 1. Introduction

Short circuit gas metal arc welding (S-GMAW) has various advantages such as low heat input, small heating area, and high thermal stability. Benefiting from advances in digital control technology, the power sources can control the voltage and current and output specific shapes of the arc curve which aim to handle the molten material transfer and control the spatter. Kah [1] proposed a classification of control techniques for S-GMAW: Natural metal transfer, current controlled dip transfer, and controlled wire feed short circuit mode. On account of the controlled wire feed short circuit mode represented by cold metal transfer welding (CMT) introducing external mechanical forces on the wire it will not be discussed here. Different shapes of the current and voltage waveform change the droplet transition, which leads to the different weld pool behavior, weld shape and microstructure.

Currently, there are a variety of S-GMAW waveform control methods on the market, which can be divided into two types. The first method is represented by Surface Tension Transfer (STT) [2] and Cold Arc process [3]. This type of methods reduces the circuit current at the beginning and end of the short circuit period to permit a smooth touch and break of the bridge of the molten metal, preventing spatter. The other type is represented by Cold MIG (Metal-Inert Gas Welding) process [1] and Low Spatter

Control (LSC) [4]. In this type of methods, considerable increase of the current gradient to accelerate the droplet detachment during the short-circuit period meanwhile the short circuit period, dramatically reduces the short-circuit period. The current in this period is reduced and occurs faster compared to a conventional short arc. However, the welding circuit of this process maintains partial current when the bridge breaks compared with the first type processes. Figure 1 compares the waveform of the conventional, the Cold Arc, and the LSC.

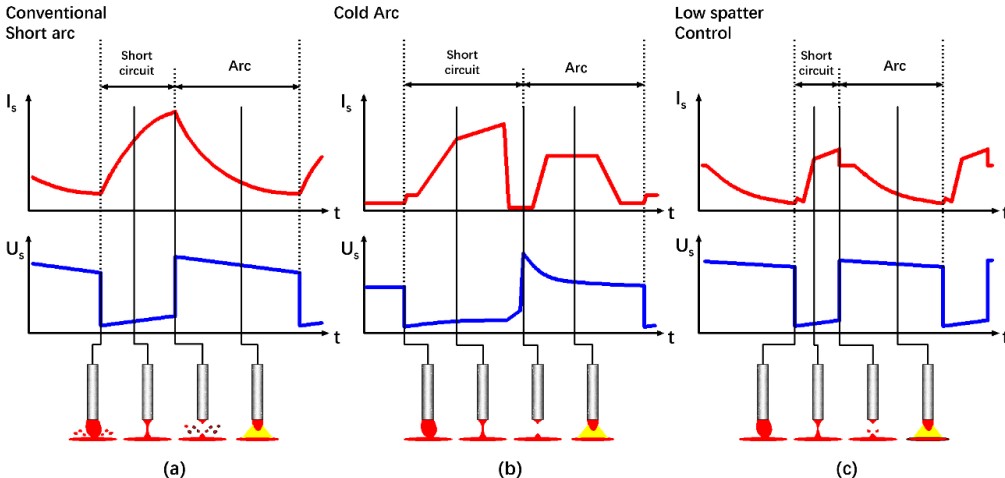

**Figure 1.** Comparison of (**a**) conventional, (**b**) Cold Arc, and (**c**) Low Spatter Control (LSC) waveform.

The behavior of weld pool is a direct reaction of weld pool state, the dynamic variation of weld pool has a great influence on the weld bead shape and microstructure. Weld pool behavior may contain sufficient information to understand the mechanisms of welding bead formation and control the stability of welding process [5–7], so it is of great significance to study the weld pool behavior of S-GMAW using different waveforms.

Many researchers have studied the weld pool behaviors in recent years. In gas tungsten arc welding (GTAW), for acquiring the amplitude and oscillation frequency of the weld pool, Yu Shi et al. [8] used line laser to illuminate the surface of weld pool. It is found that the oscillation frequency and amplitude of GTAW pool change abruptly in the process of partial penetration to full penetration. Liu [9] investigated the pulse frequency on fluid flow behavior of the weld pool in pulsed current GTAW. The result showed that weld pool oscillations triggered by pulse current lead to more heterogeneous nucleation sites, and the resonance between the movement of the weld pool and pulse current frequency greatly promotes grain refinement.

Compared with GTAW, there are complex interactions among arc plasma, droplet transfer, and pool behavior in GMAW. Richardson et al. [10] found that current pulses could not be used to trigger weld pool oscillation effectively for GMAW, the interactions between the transferred droplets and the weld pool can trigger the weld pool into oscillation. Tang et al. [11] developed a filter-reflection observation system to acquire the weld pool profile during double-pulsed gas metal arc welding process. It was found that the weld pool oscillation caused by low frequency pulse can effectively reduce the porosity and refine the weld structure.

To date, most investigations of weld pool behavior mainly focuses on GTAW and pulsed GMAW processes. However, no much research has been done in the area of weld pool characteristics with different S-GMAW waveform control methods. In this paper, a high speed photograph-images analysis system for weld pool observation was formed to capture the dynamic behavior of S-GMAW weld pool with the aim to reveal the internal relationship among the S-GMAW current waveform, the behavior of weld pool, the geometry of weld bead, and the microstructure.

## 2. Materials and Methods

### 2.1. Experimental System

The experimental system consisted of welding system, high-speed photography system, and welding electrical signal synchronous acquisition system, as shown in Figure 2.

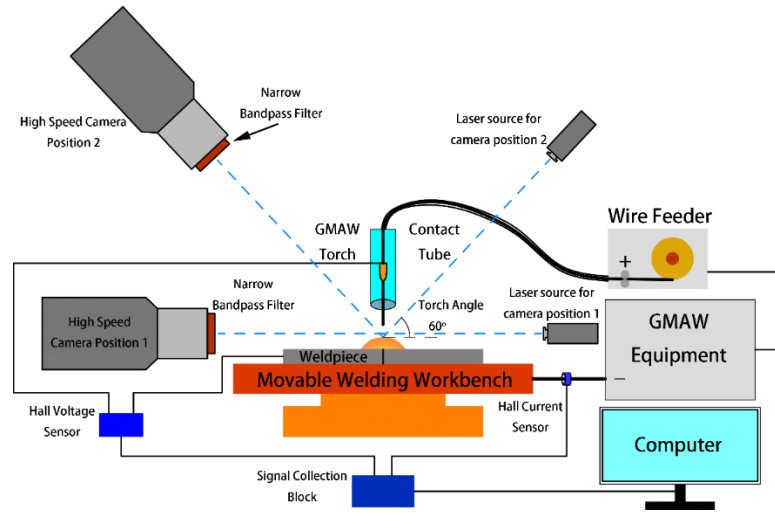

**Figure 2.** Schematic of high-speed photography system for weld pool observation.

EWM Phoenix 521 (EWM Hightec Welding GmbH, Mundersbach, Germany), EWM Cold Arc (EWM Hightec Welding GmbH, Mundersbach, Germany), and Fronius TPS5000 (FRONIUS, Pettenbach, Austria) were selected as power sources of welding system to provide needed waveforms. The welding position was PA(Flat position, as per ISO 6947). The movement of the workbench was controlled by servo motor. Current sensor, voltage sensor and signal acquisition card constituted the welding electrical signal synchronous acquisition system. Acquisition frequency of signal acquisition card is $1.5 \times 10^6$ Hz. The high-speed photography system was used to recorded the behavior of the weld pool.

In order to capture the side-view of welding pool during the welding process, the high speed photography camera was at position one of Figure 2, which is on the same horizontal plane as the welding test plate. Dynamic information about the weld pool oscillation from the high speed photography pictures was obtained by tracing the pool surface as a function of time. The shooting angle is perpendicular to the welding seam in the same plane. A light-emitting-diode (LED) was used as its excitation light source at position one, whose wave length was 850 nm, and continual output was 3 W. A laser source 850 nm near infrared filter was used to filter out strong arc during welding process. Acquisition frequency of position one is 10,000 Hz. The shooting picture is shown in Figure 3a. Positions of the camera and backlight source need to be changed to capture the contour and the flow behavior of the weld pool surface, as shown in position two of Figure 2. Different narrow-band filters were selected to obtain different information of the weld pool surface which have different spectral characteristics. The high speed camera was equipped with 850 nm near infrared filter to obtain the contour information of the weld pool with the laser shined by the same type laser source mentioned above at position two, as shown in Figure 3b. 650 nm near infrared filter was installed to obtain metal flow information of the weld pool with no laser shined, as shown in Figure 3c. Acquisition frequency of position two of Figure 2 was 2000 Hz. During video capturing, the camera was placed at an angle deviation of about 0–5°. In the analysis, the effect of these angles was not taken into account. However, based on the geometry employed, it is estimated that average errors are of the order of 1.5%.

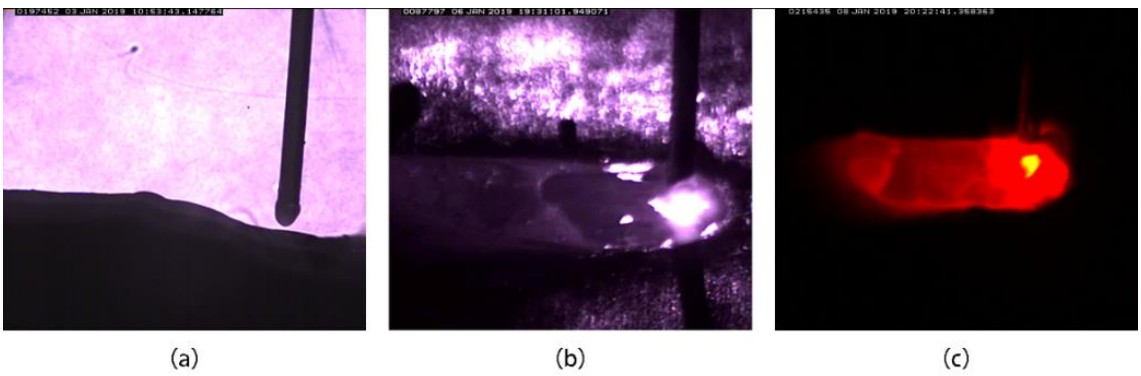

**Figure 3.** Pictures captured for different condition: (**a**) side-view shooting picture; (**b**) overhead shooting effect (850 nm near infrared filter was added); and (**c**) overhead shooting effect (650 nm near infrared filter was added).

### 2.2. Materials and Welding Parameters

In order to acquire both partly and fully penetrated weld pools under the same welding parameters, 2 mm and 4 mm of size 150 mm by 150 mm S235JR (1.0038) steel plate were selected as the base material, the weld bead was located in the middle of the plate. The length of the weld bead was 120 mm. The filler wire of ER70S-6 (G42) mild steel with a diameter of 1.2 mm was used for welding. The chemical composition of the base material and filler wire are given in Table 1. A mixture of 82% Ar + 18% $CO_2$ was used as a shielding gas, with a flow rate of 15 L/min. The travel speed was kept constant at 22 cm/min. The contact tip to workpiece distance (CTWD) was 20 mm. Welding conditions were selected that give an almost constant arc length, with an average voltage of approximately 20 V. The joint type was bead-on-plate. Welding parameters are listed in Table 2.

**Table 1.** Chemical compositions (in wt%) of base metal and filler wire (Fe balance).

| Materials | C | Mn | Si | P | S | Ni | Cr | Mo | V | Other |
|-----------|---|-----|-----|-------|-------|------|------|------|------|---------|
| S235JR | 0.17 | 1.40 | 0.3 | 0.035 | 0.035 | - | - | - | - | N 0.012 |
| ER70S-6 | 0.06–0.15 | 1.40–1.85 | 0.80–1.15 | 0.025 | 0.035 | 0.15 | 0.15 | 0.15 | 0.03 | Cu 0.5 |

**Table 2.** Welding parameters.

| No. | Waveform | Wire Feed Rate (m/min) | Voltage(V) | Thickness (mm) | Penetration |
|-----|--------------|------------------------|------------|----------------|-------------|
| 1 | Conventional | 2.4, 2.7, 3.0, 3.3 | 19 | 4 | Partial |
| 2 | LSC | 2.4, 2.7, 3.0, 3.3 | 19 | 4 | Partial |
| 3 | Cold Arc | 2.4, 2.7, 3.0, 3.3 | 19 | 4 | Partial |
| 4 | Conventional | 3.0 | 19 | 2 | Full |
| 5 | LSC | 3.0 | 19 | 2 | Full |
| 6 | Cold Arc | 3.0 | 19 | 2 | Full |

### 2.3. Principle of Measurement

The image analysis was carried out using a computer program built with LabView to obtain the change of the height of weld pool surface and the diameter of liquid bridge neck. Direct information about the weld pool oscillation from the high speed photography pictures was obtained by tracing the height of reference point on the weld pool surface as a function of time. The processing processes are as follows: (1) the contour of weld pool and wire silhouette in high speed photography was extracted by image processing system to obtain the pixel coordinates of contour, as shown in Figure 4b. (2) Direct information about the weld pool motion from the high speed video pictures was obtained by tracing the pool surface as a function of time. For this purpose, a reference point was defined on the weld pool surface, as depicted in Figure 4b. The distance between the reference point and the center of wire was

1.8 mm (1.5 times wire diameter). The Y-coordinate of reference point was measured as a function of time. In this way the change of the position of the reference point during welding can be outlined and the trend of the pool motion can be revealed [10]. (3) The shortest distance between the white line and the red line was calculated to obtain the diameter of liquid bridge necking of liquid bridge during short-circuit period. The process flow is shown in Figure 4.

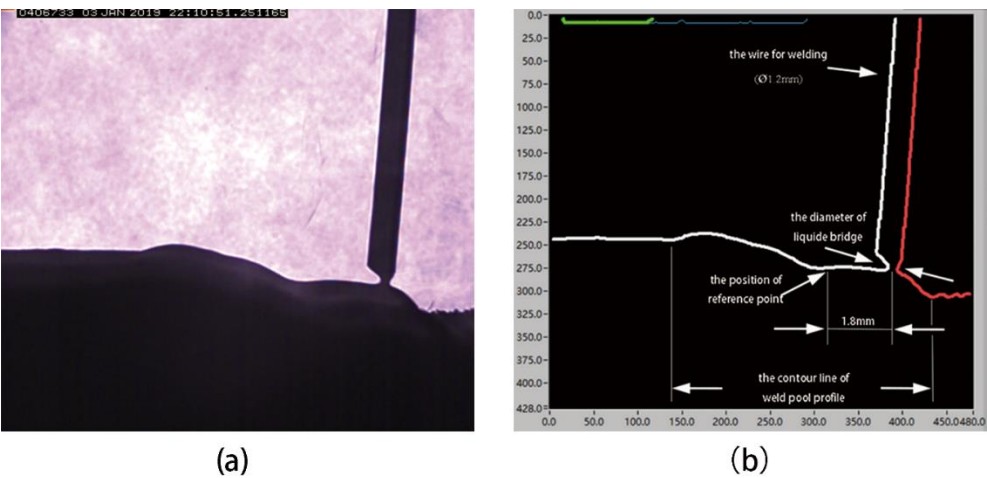

(a)　　　　　　　　　　　　　　　　　　　　　　(b)

**Figure 4.** Principle of measurement: (**a**) original images; (**b**) extraction of the outline of weld pool and the smallest diameter of liquid bridge.

## 3. Results and Discussion

### 3.1. The Metal Transfer Process and Impact on the Weld Pool

During the short circuit period, the heat source of weld pool is mainly the resistance heat of the filler material and the molten filler material. Arc heat is the main heat source during the arc period. In order to compare the heating power of different waveform control methods to the weld pool, Equation (1) was used to calculate the welding power for all waveform control methods, Equations (2) and (3) were used to calculate the welding line energy on the base metal [12,13].

$$P_w = \frac{1}{t_a + t_s}\left[\left(\int_0^{t_a} u(t)i(t)dt\right) + \left(\int_0^{t_s} u(t)i(t)dt\right)\right] \tag{1}$$

$$Q_b = P_w * \eta_{\text{eff}} * (t_a + t_s) = P_w * \eta_{\text{eff}} * t_w \tag{2}$$

$$Q_{pl} = \frac{Q_b}{v * t_w} = \frac{P_w * \eta_{\text{eff}}}{v} \tag{3}$$

where $u(t)$ is the voltage curve during welding, $i(t)$ is the current curve, $t_w$ is the welding time, $t_{arc}$ is the burning-arc time, $t_s$ is the arc-shorting time, $P_w$ is the welding power during single a droplet transfer cycle, $Q_b$ is the heat in the base material, $\eta_{\text{eff}}$ is the thermal efficiency of the welding process, $Q_{pl}$ is the heat power applied to the weld pool per unit length, and $v$ is the welding speed.

The arc length of S-GMAW is short hence the heat losses to the surrounding atmosphere are low. The effective thermal efficiency is high and the $\eta_{\text{eff}}$ of S-GMAW is 0.85 [14] which is higher than that of Pulsed GMAW and Spray GMAW.

Figure 5 shows the combination of voltage and current waveform and metal transfer process of different S-GMAW processes. In order to ensure the comparability of waveforms, the volume of the drops was similar at the time of waveforms acquisition. The arc current curve of traditional S-GMAW process is influenced by two factors: Inductance of the welding circuit and re-striking current. The re-striking current determines the peak current in the arc period. Then the current declined to the background current, this period was $t_{arc1}$ as shown in Figure 5a. Inductance of the welding circuit

determined the rate of current decline. The LSC process maintained the large current for a defined short period of time after the arc ignites to ensure that the arc had sufficient energy to heat the welding wire and the base material. Then the current decreased to the background current by the current control to regulate and initiate the next detachment, this period was $t_{arc1}$ as shown in Figure 5b. As for Cold Arc process, the current was decreased dramatically to permit a smooth break of the bridge of the molten metal at the end of short-circuit period. After the arc had been stabilized, the current was raised for a defined short period of time, known as melt pulse, to heat the welding wire and the base material. Then the current decreased to the background current, this period was $t_{arc1}$ as shown in Figure 5c. The average heating power to the base material and weld pool outlines of three waveforms is shown in Table 3.

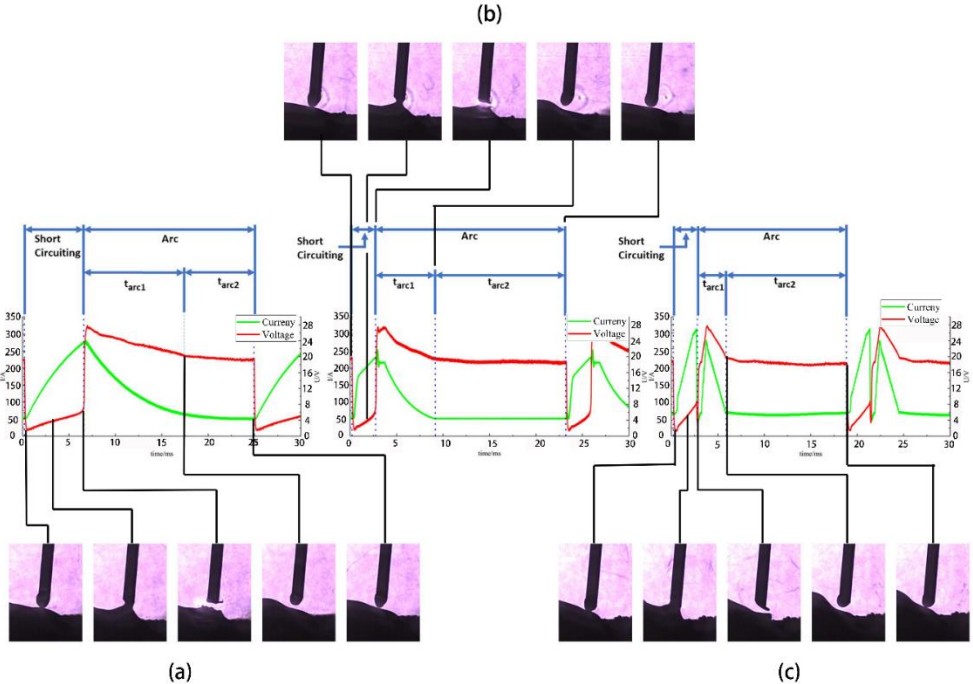

**Figure 5.** The droplet transition process: (**a**) conventional process; (**b**) LSC; and (**c**) Cold Arc.

**Table 3.** The effective heating power to the base material and weld pool outlines.

| Waveforms | Plate Thickness/mm | Wire Feed Rate/m·min$^{-1}$ | Effective Average Heating Power/KJ·m$^{-1}$ | Pool Width/mm | Pool Length/mm |
|---|---|---|---|---|---|
| Conventional | 4 | 3 | 409.915 | 5.6 ± 0.5 | 11.2 ± 1 |
| LSC | 4 | 3 | 344.656 | 5.5 ± 0.5 | 10.5 ± 1 |
| Cold Arc | 4 | 3 | 327.533 | 5.3 ± 0.5 | 9.6 ± 1 |

Figure 6 shows the profile of the weld pool during the transition period of a single molten drop. The first column of Figure 6 is the surface profiles of the weld pools at the short circuit stage, the second at the time when the liquid bridge exploded, the third at the arc stage, and the fourth at the short circuit stage of next droplet transition stage. The weld pools size was shown in Table 2. The area of the weld pool was measured by the Photoshop software, the border between the liquid and the solid was outlined manually, which could not be found by the software for tiny gray scale differences. The pool size can be obtained imprecisely by measuring the image, but the influence trend of waveform control mode on the pool size can be obtained under the same shooting condition. The results show that there was no obvious difference in the width of the weld pool, but there was a great difference in the length of the weld pool. These differences were directly related to the impact of electrical explosion at the end of short circuit.

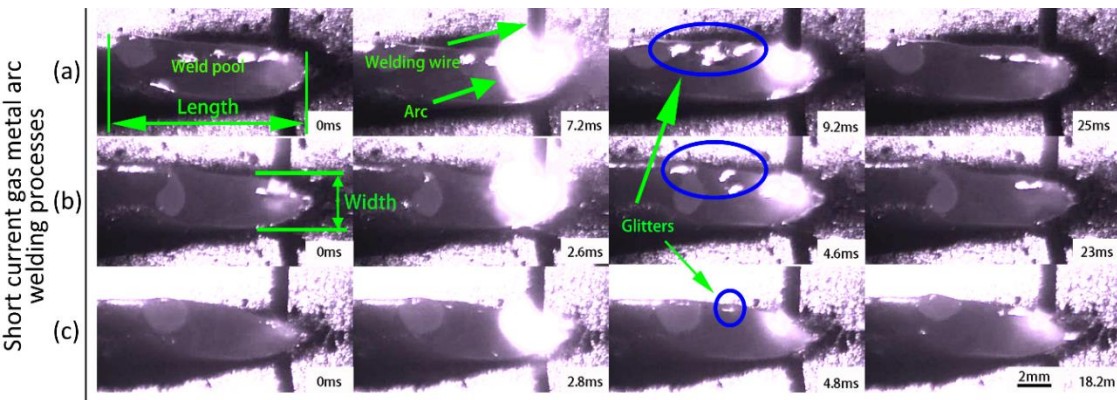

**Figure 6.** Variation of weld pool profile of short circuit gas metal arc welding (S-GMAW) under different waveforms (wire feed rate: 3 m/min, thickness of base plates: 4 mm): (**a**) conventional waveform, (**b**) LSC waveform, and (**c**) Cold Arc waveform.

As shown in Figure 6, strong arc light appeared at the moment of electric explosion. There was no obvious change in the size of the weld pool before and after the electric explosion, but there was obvious difference between the surface of different weld pools. The glitters in the blue circles of Figure 6 were due to backlight and weld pool surface, which was the mirror-like reflection. The weld pool fluctuation resulted in the change of surface curvature. The more violent the surface fluctuation of the weld pool, the greater the chance of mirror reflection and the more glitters there were. The surface of the weld pool with traditional waveform fluctuated the most, which was followed by LSC, and the weld pool of Cold Arc basically did not change. In the transition period of a single melt droplet, the energy carried by electric explosion was mainly propagated to the melt pool in the form of momentum, which changes the flow state of the metal inside the weld pool.

The resistance heat is the main factor that causes the liquid bridge explosion during short circuit period. Due to the highest resistance at the neck of the liquid bridge, it was the location where the electric explosion occurred. The instantaneous heat generation power per unit volume of the metal at the neck of the liquid bridge is calculated, the process is shown as follows:

$$R = \frac{\rho * dl}{\pi r^2} \tag{4}$$

$$P_h = I^2 * R = I^2 * (\rho * dl) / (\pi r^2) \tag{5}$$

$$P_v = \frac{P_h}{V} = \frac{I^2 * \frac{\rho * dl}{\pi r^2}}{\pi r^2 * dl} = \frac{I^2 * \rho}{\pi^2 * r^4} \tag{6}$$

where $R$ is the resistance at liquid bridge neck, $\rho$ is the resistivity of metal at liquid bridge, $r$ is the radius of liquid bridge neck which was extracted by image processing system which was mentioned above, $dl$ is the fluid bridge neck differential length, $P_h$ is the thermal power of resistance at neck of liquid bridge, and $P_v$ is the instantaneous power density of the liquid bridge. The electrical explosion is caused by overheating of the metal at the neck of the bridge. The diameter of the liquid bridge changes gently in a small area near the neck constriction whose volume can be replaced by a cylinder whose diameter is equal with the diameter of the neck of the liquid bridge. $V$ is the differential volume of the fluid bridge neck length.

The image processing system was used to extract the diameter of the shrinking neck of the liquid bridge in the short circuit period. Figure 7 are the relationship curves that show the diameter of the neck of the liquid bridge along with time.

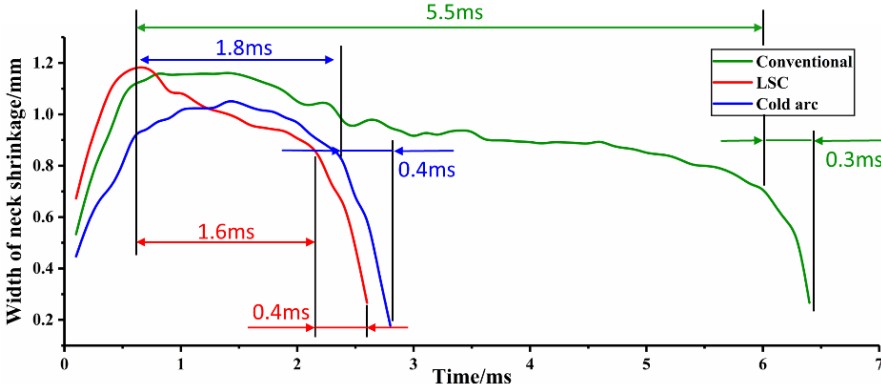

**Figure 7.** The diameter of the neck of the liquid bridge as a function of time.

It has been pointed out that the surface tension and electromagnetic pinch force are the main forces to make droplet transfer which have close relation with formation, destabilization, and break-up of short circuit liquid bridge. The curves in Figure 7 all showed a process of rapid rise, then stability, and finally rapid decline. The rapid decline stage of diameter was the process of destabilization and break-up of liquid bridge. The sharp slumping stage of three curves lasted nearly the same time as shown in Figure 7. At that time, the current in the welding loop were 280 A, 210 A, and 50 A in conventional, LSC, and Cold Arc, respectively, as shown in Figure 5. The results showed that electromagnetic shrinkage force had little effect on the duration of destabilization and break-up of short circuit liquid bridge. The difference of stability times of liquid bridges was obvious, which indicated that the rising rate of loop current in the short circuit stage can effectively promote the formation of neck of liquid bridge and greatly reduce the short circuit stage time.

Figure 8 shows the relationship curves of the instantaneous power density of liquid bridge neck with time under three waveform conditions. As shown in Figure 8, The curve of the instantaneous power density of the liquid bridge along with time was acquired by substituting the diameter of the shrinking neck of the liquid bridge and the current corresponding to it into Equation (6).

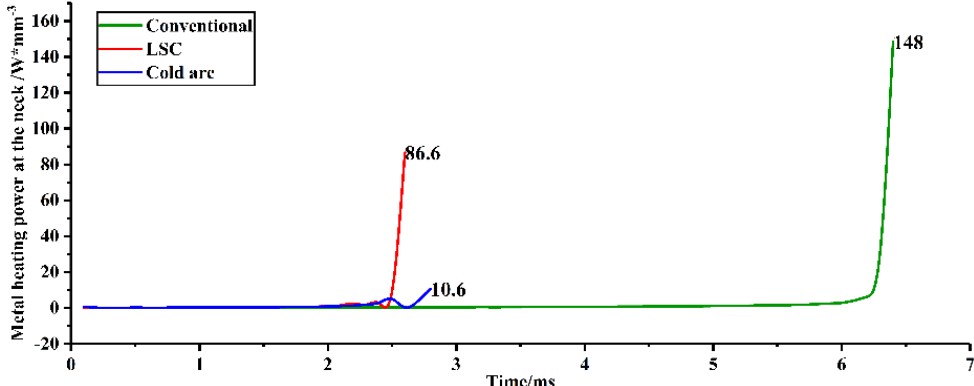

**Figure 8.** The instantaneous power density of the liquid bridge as a function of time.

It can be seen from Figure 8 that the instantaneous power density of the liquid bridge was extremely low during short circuit period for most of the time. The energy accumulated in a very short time before the liquid bridge explosion is the main factor influencing the impact of electric explosion. Therefore, the instantaneous power density of the liquid bridge during the burst can effectively measure the magnitude of the electric explosive impact force. The instantaneous power density of liquid bridge metal in cold arc power supply was relatively small. The instantaneous power density of liquid bridge metal at the end of short circuit in LSC process was about half of that of traditional process, and the electric explosion impact force was less than that of traditional process. The impact force of electric

explosion determines the dynamic characteristics of weld pool. Figure 9 shows the probability density distribution of oscillation amplitude of weld pool:

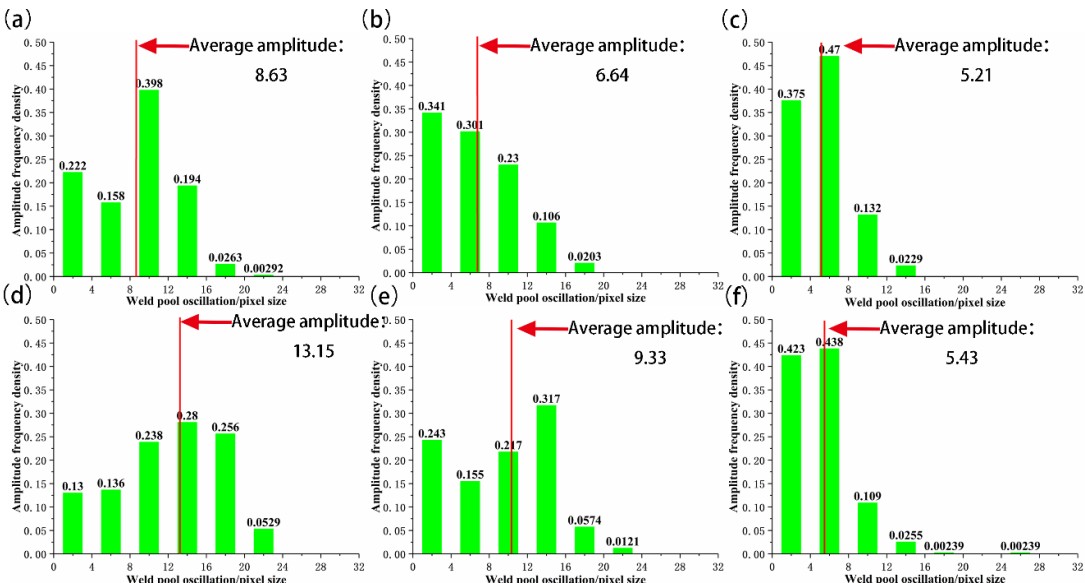

**Figure 9.** The probability density distribution of oscillation amplitude of weld pool: (**a**–**c**) are the amplitudes of the partial penetration pool of conventional process, LSC, and Cold Arc and (**d**–**f**) are the amplitudes of the full penetration pool of conventional process, LSC, and Cold Arc.

The amplitude of weld pool is proportional to the impact of electric explosion. The probability of large amplitude of weld pool in traditional process was greater than that of LSC and Cold Arc. The impact of electric explosion on the weld pool in Cold Arc process was very small, and the liquid level of the weld pool had no obvious fluctuation.

The results of the statistics of oscillation amplitude are in good agreement with Figure 6. The amplitude was proportional to the impact of surface traveling wave on the boundary of weld pool. This can explain the obvious difference in the length of weld pool with little difference in the width of weld pool. The oscillation amplitude of weld pool was affected by the state of weld pool. The full penetration pool had larger amplitude of the weld pool was affected by the state of the weld pool; compared with the partial penetration. When the weld pool was impacted, the bottom of the full weld pool was liquid metal level, which had little effect on the downward movement of metal flow. As a result, the amplitude of full penetration pool was larger than that of partial penetration pool under the same welding parameters.

### 3.2. Oscillation of Weld Pool

High speed photograph pictures showed that the liquid waves in S-GMA welding were triggered primarily by the electric explosion, not by the change in the arc pressure during the arc period [15]. The relationship curves the height of the reference point on the weld pool surface with time were obtained by using the method described in Section 2.3, as shown in Figure 10.

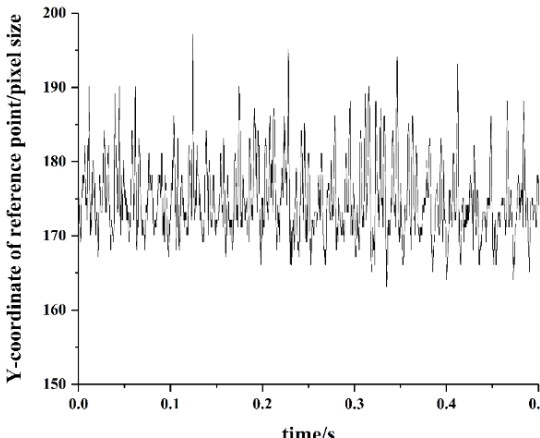

**Figure 10.** Oscillation signal extracted by high speed photograph-image analysis system.

The period of droplet transfer cycles fluctuated randomly within a range, and the oscillation curve of weld pool is a time-varying signal, which causes the oscillation frequency of weld pool change within a range. The Fast Fourier Transform Algorithm (FFT) could not extract the characteristics of weld pool oscillation. The Continuous Wavelet Transform (CWT) can analyze time-varying signals in time domain and frequency domain simultaneously. In this paper, Morlet continuous wavelet transform was applied to weld pool oscillation signals at different wire feeding speeds. Morlet wavelet base function is shown in Equation (7):

$$\psi_{a,b}(t) = \sqrt{a}exp\left(i\omega_0\frac{(t-b)}{a}\right)exp\left(-\frac{(t-b)^2}{2a^2}\right) \tag{7}$$

In the continuous wavelet transform, the scale vector $a$ is associated with the central frequency and the support interval of the basis function, and the frequency of weld pool oscillation and its corresponding time frequency resolution can be obtained at any time. For a particular scale vector, the signal frequency allowed by the wavelet transform should be close to the corresponding frequency of the scale vector. Therefore, the continuous wavelet transform can clearly reflect the variation of oscillation frequency with time. In this experiment, wavelet transform is carried out on the acquired signal of melt pool oscillation, and the center frequency $\omega_0$ of base function was three. The oscillation frequency range of traditional GMAW weld pool is below 300 Hz [16]. The scale vector $a$ selected in this experiment was between 50 and 700, and the corresponding oscillation frequency identification range was 40–600 Hz. $b$ is the duration of signal acquisition. The contour diagram of transform coefficient of signal reflects the energy density distribution of the signal in the time-scale plane. The energy of the signal is mainly concentrated around the wavelet-ridge-cure in the time-scale plane, from which the instantaneous frequency of the signal can be determined. Signal sampling frequency ($f_{\text{Sampling frequency}}$) was equal with the fps of high-speed photography, and the corresponding relationship between the oscillation frequency of weld pool ($f_{\text{Oscillation frequency}}$) and the scale vector $a$ of wavelet-ridge-cure is as Equation (8):

$$f_{\text{Oscillation frequency}} = \frac{f_{\text{Sampling frequency}} \cdot \omega_0}{a} \tag{8}$$

Figure 10 is the contour diagram of continuous wavelet transform coefficient of weld pool oscillation signal of traditional S-GMAW process under different wire feeding speeds:

It can be seen from Figure 11 that the oscillation of weld pool of S-GMAW had significant periodicity. The relationship curve of the oscillation frequency with different wire-feed speeds is shown in Figure 12. When the wire feeding speed was 2.4 m/min, the weld pool volume was small, resulting high oscillation frequency of the weld pool. With the increase of wire feeding speed, the volume of

weld pool increases, the propagation time of travelling wave on the surface of weld pool increased, and the oscillation frequency decreased.

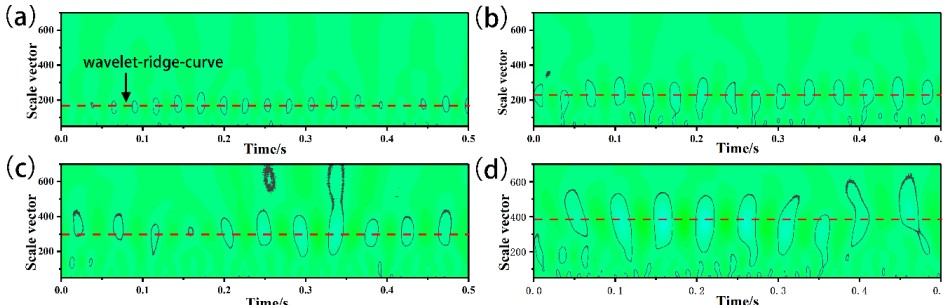

**Figure 11.** The contour diagram of transform coefficient of weld pool oscillation: (**a**) 2.4 m/min; (**b**) 2.7 m/min; (**c**) 3.0 m/min; and (**d**) 3.3 m/min.

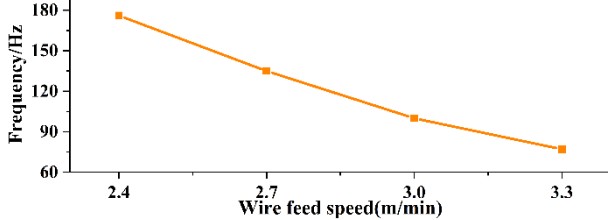

**Figure 12.** The oscillation frequency of weld pools with different wire-feed speeds.

Different waveforms and penetration states of weld pool led to different oscillation frequencies. Figure 13 shows the contour curves of the oscillation wavelet transform coefficients of the fusion and partial weld pools of three waveforms, and Table 4 shows the oscillation frequency statistics.

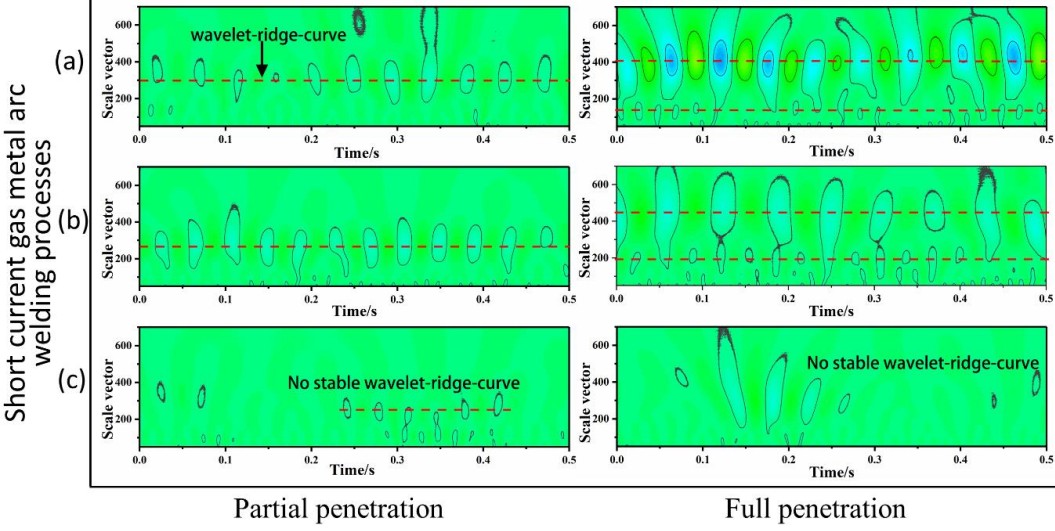

**Figure 13.** The contour curves of the oscillation wavelet transform coefficients of the fully and partly penetrated weld pools: (**a**) conventional process; (**b**) LSC; and (**c**) Cold Arc.

**Table 4.** The oscillation frequency statistics.

| Waveform | Partial Penetration | Full Penetration |
|---|---|---|
| Conventional | 100 Hz | 75 Hz<br>200 Hz |
| LSC | 112 Hz | 68 Hz<br>165 Hz |
| Cold Arc | Not available | Not available |

As can be seen from Figure 13, there was a significant difference in the oscillation frequency between the partly and fully penetrated weld pools. These pictures at both sides of Figure 13 are the contour diagram of the distribution of oscillation wavelet coefficients of partly and fully penetrated weld pool using different waveforms. Only one frequency occurred during the oscillation process of partly penetrated weld pool, while there were two characteristic frequencies on the oscillation spectrum of the fully penetrated weld pool. The difference between high frequency and low frequency was generally about 40 Hz, which indicated that there were two oscillation periods of different frequencies in the weld pool. Zacksenhouse [17] established a pool analysis model based on the stretch film theory and studied the oscillation frequency of the full penetration pool. In the full penetration pool, the vibration frequency is obviously lower than that of the partial penetration pool, and the amplitude of the oscillation of the fully penetrated weld pool is relatively larger than that of the partly penetrated weld pool due to the disappearance of the bottom constraint, which was consistent with the Figure 9.

In order to explain the two oscillation frequencies of the full penetration pool, the metal flow process in the pool should be considered, as shown in Figure 14. With the impact of electric explosion, the liquid weld pool flowed radially symmetrically with the arc axis. In the full penetration pool, axial flow occurred for the bottom of the weld pool was no longer supported by any solid material. The liquid in the middle of the weld pool can move vertically, while the liquid in the periphery of the weld pool was supported by the solid material and forced to flow laterally. It led to the fact that although the weld pool was in the state of full penetration, the traveling wave propagation process was still similar to that of non-molten penetration at the periphery of the weld pool. During travelling S-GMAW welding process, the penetration position was relatively small compared with the length of the weld pool, and most of the weld pool metal was still supported by solid metal at the bottom of the weld pool, the oscillation behavior was similar to partial penetration. Therefore, the full penetration pool had two characteristic oscillation frequencies: High frequency and low frequency.

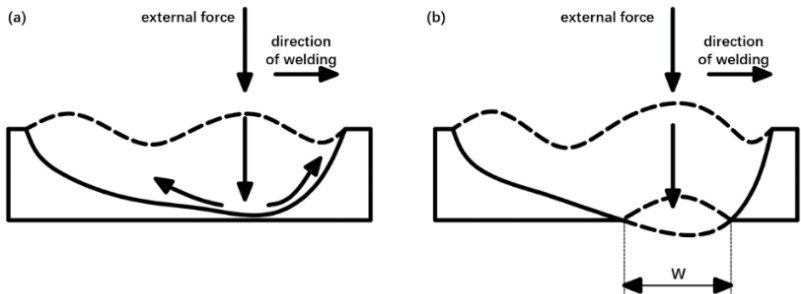

**Figure 14.** Oscillation mode of weld pool: (**a**) partial penetration; (**b**) full penetration.

The characteristics of the pool oscillation of three waveforms were different. As shown in Figure 13, No stable wavelet-ridge-cure occurred in the contour curves of the oscillation wavelet transform coefficients of Cold Arc, so the weld pools of Cold Arc had no stable oscillation frequency. The Cold Arc process reduced the current at the end of short-circuit stage, which greatly reduces the impact of electric explosion on the pool. At the same wire feeding speed, the weld pool oscillation frequency of LSC was slightly lower than that of conventional process, which was related to the size of weld

pool and the electric explosive impact force at the end of short circuit, and the surface tension of weld pool was one of the factors that caused the difference of the frequency. Different waveforms led to different surface temperature of the weld pool, resulting in different surface tension of the weld pool. The surface tension of the weld pool metal is also one of the important factors affecting the oscillation frequency of the weld pool.

### 3.3. Flow Behavior of Weld Pool

To study the weld pool flow behavior, positions of the camera and backlight source need to be changed, as shown in position two of Figure 2. 650 nm near infrared filter were installed to obtain metal flow information of the weld pool. When active gas served as a shielding gas, alloying elements like silicon and manganese, which were present in the base metal and the wire, had a high affinity to react with oxygen and form silicon oxide and manganese oxide. These oxides accumulate on the surface of the weld pool and form slag [18]. the slags have a lower density than the molten metal and follow the flow pattern of the weld pool. Hence, slag flow pattern and accumulation location can disclose the weld pool flow behavior [19].

GMAW weld pool consists of the hot part of the weld pool and the cold part of the weld pool [19,20]. The hot part of the weld pool consisted of the area directly under the arc and the surrounding region, and the cold part of the weld pool is located behind the hot part of weld pool. According to Grong et al., the metal oxides in the high temperature zone of the weld pool exists in the form of metal oxide powder, which cannot aggregate into slags [20,21]. The metal oxides in the cold part of weld pool accumulate into blocks to form slags. Slag is a poor conductor of heat and prevents the red glow of the weld pool, which can block the light at a wavelength of 650 nm. The slag flow pattern can be clearly observed by using 650 nm polaroid as filter.

Figure 15 represented the frames from the high speed video to show the partly penetrated weld pool flow pattern and the slag accumulation location for conventional process, LSC, and Cold Arc, respectively. The white powder in the front and middle of the weld pool was the silicon oxide and manganese oxide particles, which are separated from the weld metal due to the strong turbulence in the weld pool in this part and pushed to the low temperature area of the weld pool under the action of the pool flow. The slags flow patterns behaviors of different processes showed significant difference.

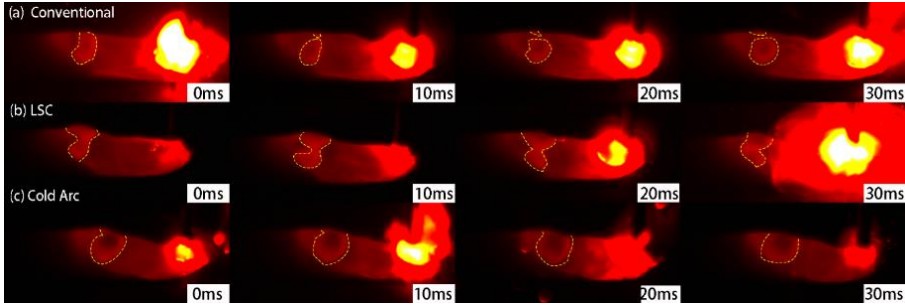

**Figure 15.** The partly penetrated weld pool flow pattern and the slag accumulation location: (**a**) Conventional; (**b**) LSC; (**c**) Cold Arc (the slag islands were outlined by white dotted lines).

The weld pool flow in partly the penetrated weld pool is explained with the assistance of Figure 16. When the weld pool is forced to flow downwards (by the external force), it is blocked by the base metal and is forced to flow to the back of the weld pool. At the back of the weld pool, the metal liquid flow will rebound off the solid metal interface and flow to the front of the pool. By the present experiment results in Section 3.1, the impact of the traditional S-GMAW process on the weld pool is the strongest of the three, and the bounced metal flow was the strongest which led to the formation of two spinning large slag islands. The impact of the LSC on the weld pool was smaller than the conventional process. The dumbbell shaped slag island in LSC was formed with co-extrusion of the bounced metal flow and the backward flow of metal on the surface of weld pool. Cold Arc process has little impact on the weld

pool, because the bounced metal strength was negligible, so the metal oxides gathered into a single round island of slag.

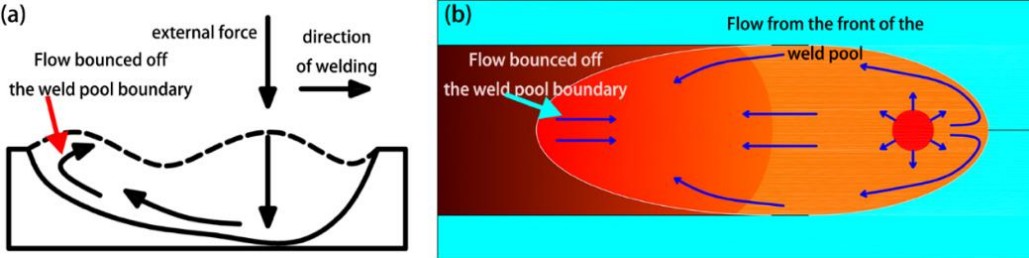

**Figure 16.** Schematic diagram of flow of partial penetrated weld pool: (**a**) inside. (**b**) surface.

As shown in Figure 17, compared with partial penetrated weld pool, the characteristics of the full penetrated weld pool flow pattern have the obvious differentiation, the full penetrated weld pools also have obvious low temperature zone and high temperature zone, but the metal oxides in the cold part of the weld pool did not gather and form huge slag islands, instead distributed at the back of the weld pool evenly. The bottom surface of the full penetrated weld pool can expand and contract with impact, and the energy was absorbed due to the existence of the free surface at the bottom, as shown in Figure 18. The liquid metal in the hot part of weld pool was not pressed, so the volume and length of the fully penetrated weld pool was larger than that of partly penetrated weld pool. For the fully penetrated weld pool, the slag at the end of the pool formed discrete scattered islands and did not gather into a large slag island, as was observed for the partial penetration case. This change was simply dependent on the surface wave of the weld pool.

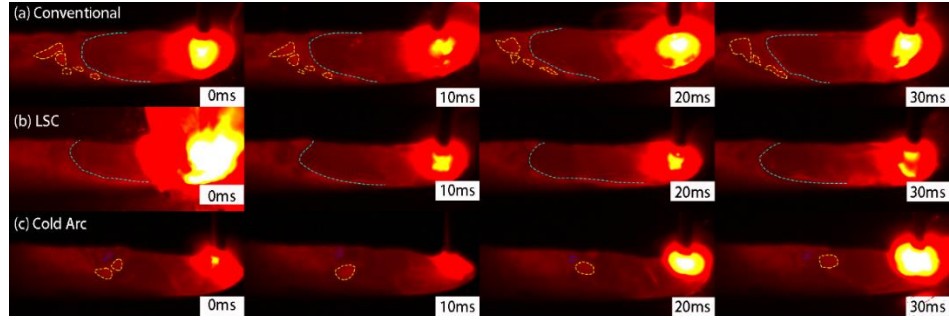

**Figure 17.** The fully penetrated weld pool flow pattern and the slag accumulation location: (**a**) Conventional; (**b**) LSC; (**c**) Cold Arc (the slags were outlined by dotted lines).

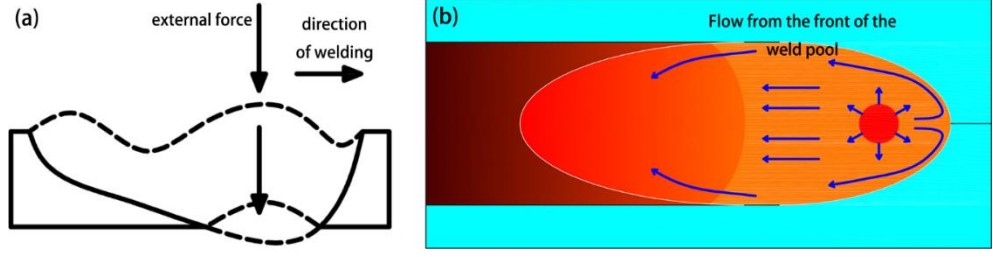

**Figure 18.** Schematic diagram of flow of fully penetrated weld pool: (**a**) inside. (**b**) surface.

Since the composition of shielding gas was consistent, the influence of the Marangoni flow can be excluded. And for the cold part of the weld pool lie on the further from the center of the arc, the influence of plasma flow force and electromagnetic force can be ignored. The difference observed in weld pool flow pattern was attributed to the varied degrees-of-freedom of weld pool.

### 3.4. Geometry and Microstructure of Weld Bead

The weld geometry often qualifies melting characteristics of base metal, amount of weld deposition, welding heat input, energy distribution, and regulation of the flow of liquid metal to control its shape at various parameters [6]. The energy distribution and the metal transfer in the welding process depend on the waveforms of S-GMAW, which were both the main factors that affects the fluidity of the weld pool, and further affects the weld microstructure.

#### 3.4.1. Geometry of Weld Bead

The change of waveforms had remarkable influence on weld geometry. It is of great significance to study the weld bead geometry of S-GMAW under different wave forms. Transverse sections are shown in Figure 19 for beads on plate deposition which contains partial penetration (the thickness of base material was 4 mm) as well as fully penetration (the thickness of base material was 2 mm), respectively.

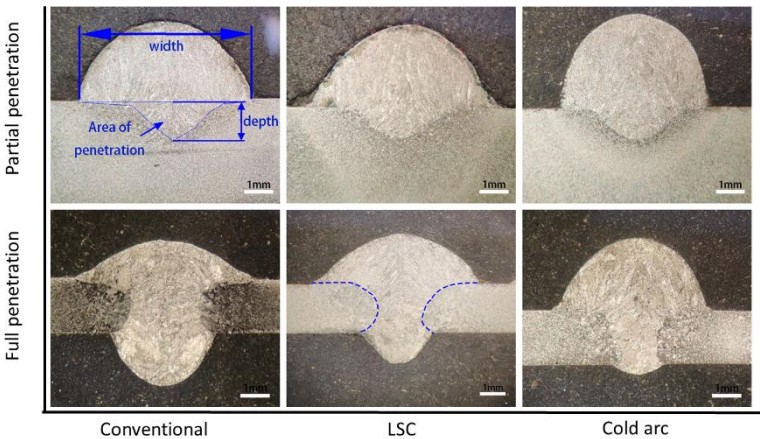

**Figure 19.** The geometry of the weld beads with different waveforms (wire feed rate: 3 m/min, voltage: 19 V).

As shown in Figure 19, the area of base metal fusion of Cold Arc was smaller than that of the other two waveforms. As mentioned in Section 3.1, the average heating powers of LSC and Cold Arc to the base material were similar. The difference of weld pool behaviors was the main factor caused the difference of the area of bead deposit.

The oscillation intensified the convection of the weld pool, which led to the metal in the hot part to flow to the pool boundary, which intensified the melting of the base material. The impact of the conventional process and LSC on the weld pool was much larger than that of Cold Arc, so the area of base metal fusion for the Cold Arc was the smallest. Due to the oscillation amplitude of the fully penetrated pool was larger than that of the partly penetrated pool, as well as the bad heat dissipation of thin plate, the area of full penetration base metal fusion was much larger than that of the partial penetration.

The influence of electric explosion on the weld pool changed the depth and width of weld pool. Figure 20 listed the variation of weld geometry with increasing of wire feed rate. Due to the narrow range of weld parameters of the full penetration pool, this paper only listed the weld forming parameters of the partial penetration pool with different wire feed rate.

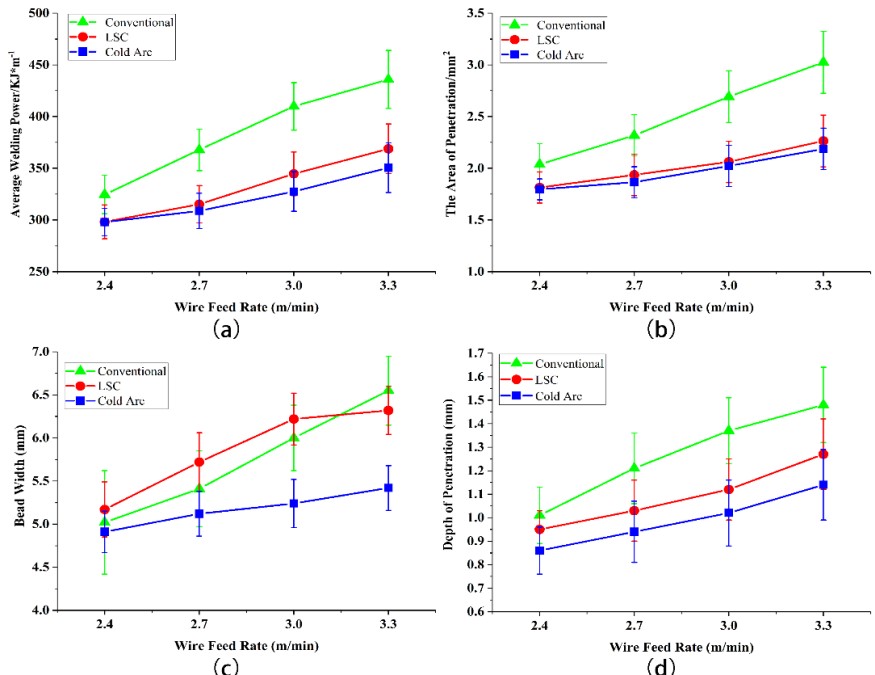

**Figure 20.** Effect of wire feed rate on geometry of fusion (partial penetration) in bead on plate deposition of S235JR: (**a**) average welding power; (**b**) area of penetration; (**c**) width; and (**d**) depth.

As shown in Figure 20, the main factor that determined weld geometry was welding heat input. The influence of weld pool behavior on weld geometry and penetration area was not obvious. Low wire feeding speed resulted in smaller peak current, lower energy input difference, lower effect of electric explosion impact on the pool, and more consistent weld formation of three waveforms. The difference of weld forming was more obvious with the increase of wire feeding speed.

### 3.4.2. Microstructure of Weld Metals

The waveform of S-GMAW has great influence on the microstructure of weld bead. It determines the dynamic behavior of the weld pool, which further influences the solidification behavior of the weld pool. In a large part, solidification behavior determines microstructure of weld metal. This microstructure of weld metals is probably due to the complex interactions between weld thermal cycle, cooling rate, and the prior austenite grain size [22]. In this study, the weld metal composition remained almost constant under the conditions that the same base metal, filler metal, and shielding gas were used in all experiments, therefore, the change of microstructure is related to the change of arc heat input and weld pool behavior.

For the same welding condition, the weld pool thermal behaviors between the three waveform processes were completely different, which resulting in obvious and different weld microstructure, as shown in Figures 21 and 22. The proeutectoid ferrite (PF) volume fraction precipitated at the initial austenite grain boundary were related to the welding heat cycle. The lower the cooling rate of weld bead, the greater the amount of PF formation. Poor heat dissipation can aggravate this phenomenon, as the full penetration weld was prepared with 2 mm steel plate, which inhibited the heat transfer and reduced the cooling rate, it caused thicker PF compared with the partly penetrated weld pool.

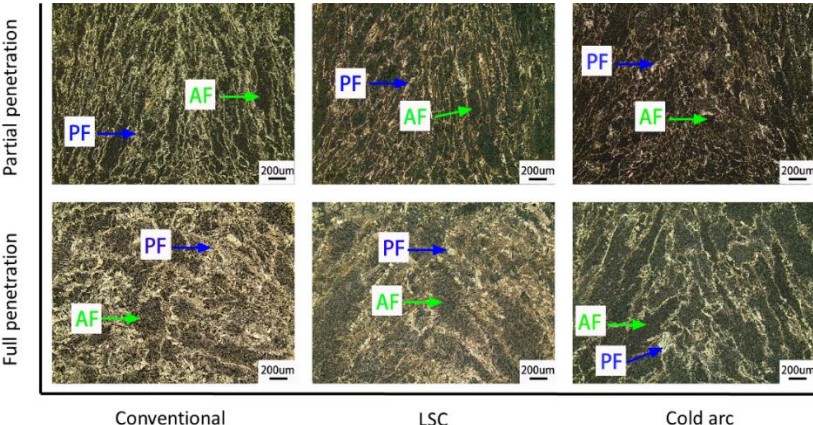

**Figure 21.** The microstructures of the weld beads with different waveforms (magnification: 50 times).

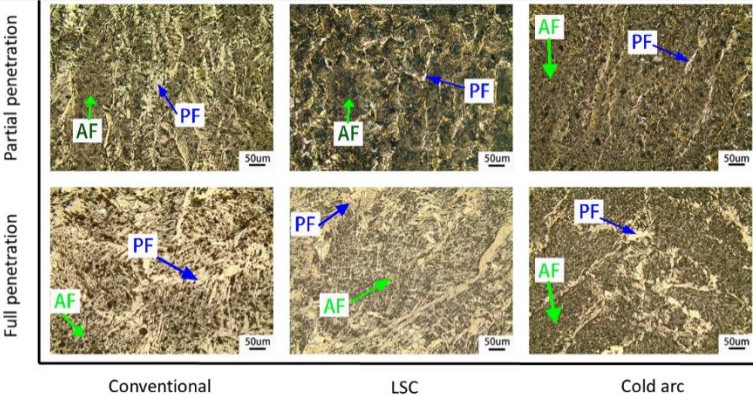

**Figure 22.** The microstructures of the weld beads with different waveforms (magnification: 200 times).

The heat input of conventional process was larger than the LSC and the Cold Arc, resulting in larger PF microstructure. The heat input of LSC was close to Cold Arc, but the oscillating behavior of the weld pool aggravated the heat exchange between the metal liquid in the high temperature zone and the low temperature zone, resulting in the slow cooling rate and the precipitation of more PF in the initial austenite grain boundary.

PF usually precipitates at the initial austenite grain boundary, and the distribution of PF reflected the initial austenite grain boundary [23]. The directivity of PF structure in partly penetrated weld bead was obvious, which indicated that most of the initial austenite grain were columnar. In the fully penetrated weld bead of conventional process and LSC, the structure of PF microstructure no longer had direction, which indicated that the solidification and growth process of initial austenite grains could be changed by the oscillation of the weld pool.

In the weld microstructure, acicular ferrite (AF) can effectively increase the mechanical properties (particularly toughness) of the weld. Therefore, the volume fraction of acicular ferrite in Figure 21 was measured, image processing software was used to measure the area fraction of different phases in the Figure. The area fraction of AF and PF phases in the metallographic picture were approximately equal to their respective volume fractions, and the results were shown in Figure 23. The results show that the volume fractions of AF were significantly different, due to the change of the weld heat gradient within the different waveforms. The heat gradient of the weld pool of the fully penetrated weld bead was smaller than that of the partial penetration, which resulted in a decrease of the volume fraction of AF. The weld pool oscillated continuously in the traditional process and LSC, the weld pool was agitated and the thermal gradient declined, resulting in the decrease of AF content. The AF content in the weld microstructure of the traditional process and LSC was lower than that of Cold Arc.

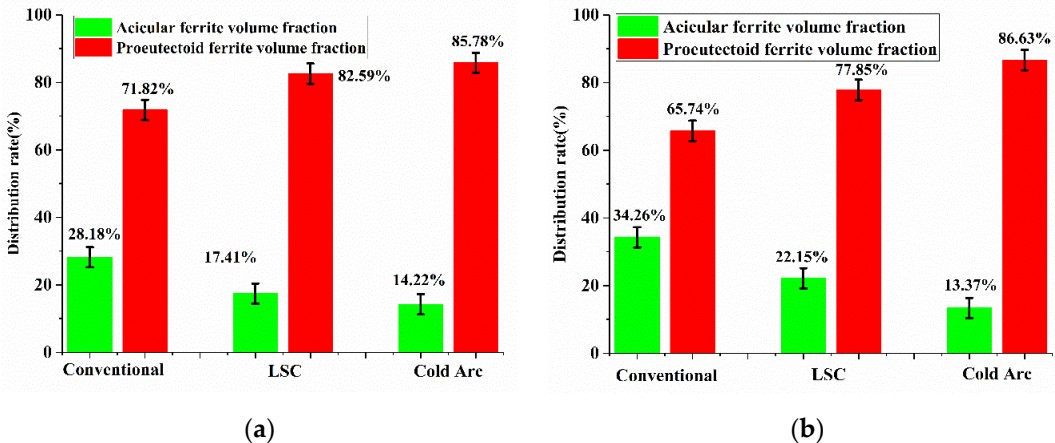

(**a**)  (**b**)

**Figure 23.** Acicular ferrite and proeutectoid ferrite volume fraction in weld deposits: (**a**) partial penetration; (**b**) full penetration.

The AF microstructures of different bead-on-welds as shown in Figure 24 reveal different grain size. The average grain sizes of AF grain were calculated using intercept method (as per ASTM E112-10). In general, the grain size of weld metals can be typically correlated with the heat input or the cumulative effect of weld parameters. As mentioned in Table 3, the heat input of traditional process was obviously larger than that of LSC and Cold Arc, which caused the grain size of conventional process was larger than the others. The heat input of LSC was approximately equal with the Cold Arc, the size of acicular ferrite precipitated in the weld of LSC was larger than that of Cold Arc. In the case when the heat input was approximately equal, the change in AF size also verified the difference of thermal gradient of welding pools between different waveforms. The decrease of thermal gradient resulted in AF grain grow.

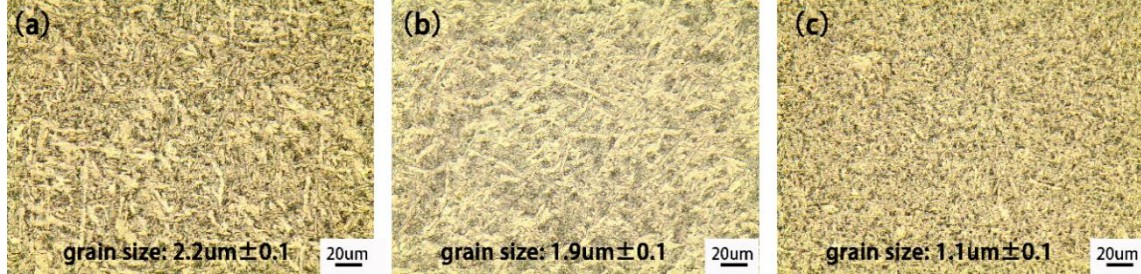

**Figure 24.** Optical micrographs of acicular ferrite (**a**) conventional, (**b**) LSC, and (**c**) cold Arc.

The weld microstructure of low carbon steel was the result of solidification and solid phase transformation of the weld pool in non-equilibrium state. The columnar-to-equiaxed transition caused by the oscillation of the weld pool had not been observed, but the oscillation homogenized the temperature distribution of the weld pool, reduced the temperature gradient of the weld pool, and resulted in the coarsening of the weld structure in the process of solid phase transformation.

## 4. Conclusions

In this paper, the behavior characteristics of S-GMAW weld pool were studied, and the differences of the pool behavior and weld microstructure were compared under different waveforms. The conclusions are as follows:

(1) In short-circuit period, the duration of destabilization and break-up of the liquid bridge is mainly related to the surface tension of the liquid metal, not the loop current. However, the rise rate of the loop current can effectively shorten the stability time of the liquid bridge and promote the

formation of the neck of the short-circuit liquid bridge. The liquid bridge explosion is related to the instantaneous power density of liquid bridge metal.

(2) The weld pool oscillation is triggered by the pressure of the electric explosion. The oscillation of the weld pool can be monitored visually by high-speed photography imaging. The oscillation of the weld pool has natural frequencies which decrease with the increase of volume of weld pool. In the case of partial penetration, only one natural oscillation frequency can be detected. In the case of full penetration two different oscillation frequencies can be detected.

(3) The shape of slag on the surface of the weld pool and the flow behavior of the weld pool can reflect the penetration state of the weld pool. The different boundary conditions between the partial and full penetration cause different flow behavior of the weld pool, which leads to the fact that the slag tends to aggregate into large blocks in partial penetration, while the slag in the fully penetrated weld pool cannot aggregate into blocks. Large slag island can be deformed or split apart with different impact strength of electrical explosions.

(4) Compared with the influence of weld heat input on the size of weld pool, the effect of weld pool oscillation is not obvious. The oscillation imparts a negative effect on the weld microstructure, along with the aggravation of the weld pool oscillation, the content and size of proeutectoid ferrite in the weld microstructure increases, the content of acicular ferrite decreases while the grain size increases.

**Author Contributions:** Methodology, T.C.; Software, T.C.; Investigation, T.C., P.Z., Writing—Original Draft Preparation, T.C.; Writing—Review and Editing, S.X., T.C., B.W.; Supervision, S.X.; Project Administration, S.X., W.L.; Funding Acquisition, S.X.

**Funding:** This work was funded by the National Natural Science Foundation of China, grant No.51675269 and the Priority Academic Program Development of Jiangsu Higher Education Institutions (PAPD).

**Conflicts of Interest:** The authors declare no conflict of interest.

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
