# Peer review of "Study on Short-Circuiting GMAW Pool Behavior and Microstructure of the Weld with Different Waveform Control Methods"

_metals, doi:10.3390/met9121326_

Round 1

Reviewer 1 Report

Dear Authors,

the article is long and confusing. Your conclusions from the literature should be critical checked by your own experiences, you results and technical possibilities.

The english is realy good only some descriptions are not clear.

Your practical work is good and some ideas for evaluation are good.

But the article has scientific inconsistencies and needs more than major revisions.

Please check the notes and rewrite and reconsider.

Sorry for that.

Author Response

Dear reviewer,

We deeply appreciate the time and effort you have spent in reviewing our manuscript. Based on your comments and suggestions, we have made modification on the manuscript and the revisions were highlighted in red in the revised manuscript. Here below are our descriptions of the revisions according to your comments.

Point 1: Page 1: Line 37: On account of the controlled wire feed short circuit mode represented by cold metal transfer welding (CMT) introducing external mechanical forces. forces on what?

Response 1: Thanks very much for your careful review of our manuscript. The external mechanical forces mentioned above is introduced on the wire. The CMT process involves a mechanically controlled forward and backward motion of welding wire to assist the droplet transfer during short-circuiting, and material transfer can occur without the aid of electromagnetic force.

Point 2: “in the current” (Page 1 Line 44) should be replaced by “of the current gradient”.

Response 2: Thanks very much for your careful review of our manuscript. We have rewritten this sentence (Page 2 Line 45).

Point 3: “- the short-circuit period, the short circuit period-” (Page 2 Line 45) should be replaced by “- the short-circuit period meanwhile the short circuit period-”

Response 3: Thanks very much for your careful review of our manuscript. We have rewritten this sentence (Page 2 Line 46).

Point 4: “and the resonance between the weld pool and pulse current frequency greatly promotes grain refinement.” (Page 2 Line 64) should be replaced by “and the resonance between the movement of the weld pool and pulse current frequency greatly promotes grain refinement.”

Response 4: Thanks very much for your careful review of our manuscript. We have rewritten this sentence (Page 2 Line 66).

Point 5: Image annotation needed to be changed (Figure 2 and Figure 4).

Response 5: Thanks very much for your careful review of our manuscript. We have redrawn Figure 2 (Line 87) and Figure 4 (Line 148), necessary annotations were added and changed.

Point 6: EWM Phoenix 521 and EWM Cold Arc both can realize standard/natural S-GMAW.

Response 6: Thanks very much for your careful review of our manuscript. Both weld power (EWM Phoenix 521 and EWM Cold Arc) can realize standard S-GMAW, while due to equipment maintenance in our laboratory, both the weld powers were used and listed.

Point 7: what is the type, power and wavelength of the laser source?

Response 7: Thanks very much for your careful review of our manuscript. We added the type, power and wavelength of the laser source. A Light-emitting-Diode (LED) was used as its excitation light source, whose wavelength was 850nm, and continual output was 3W. (Page 3 Line 100-110).

Point 8: what grade, A. B, C, D? Please specify the base and the filler material, for example, a list of chemical composition and write international material numbers.

Response 8: Thanks very much for your careful review of our manuscript. We added the grade of base-material, the international material numbers (Line 120 and 121), a list of chemical composition of base material and wire material, as shown in Table 1, Line 128.

Point 9: 20mm CTWD was normally too long, should be 15mm.

Response 9: Thanks very much for your careful review of our manuscript. In order to get a bigger perspective for high-speed photography, we read similar research papers for reference (“Yudodibroto, B, Y, B.; Hermans, M J M. et al. Observations on Droplet and Arc Behaviour during Pulsed GMAW. Weld World 2009, 53(7-8): R171-R180. ”). On the basis of ensuring the quality of welding joints, we increased CTWD to 20mm.

Point 10: The equation (1) was wrong, please recalculate power and time (multiplication of both is the energy). The heat input is definitely more than 10%.

Response 10: We have read the references you recommended. The previous calculation equation in this paper was wrong, we corrected the equation and updated the references. We rewrote section 3.1 and recalculated the weld heat power, as shown in Table 3. Thanks very much for your recommendation

Point 11: Image quality is too poor and no visible (figure 5).

Response 11: Thanks very much for your careful review of our manuscript. We have redrawn Figure 5 (Line 167) to present a clear figure and legend.

Point 12:” electrical signal waveform” (Line 152 Page 5) should be replaced by “voltage and current waveform”.

Response 12: Thanks very much for your careful review of our manuscript. We have rewritten this sentence (Line 169 Page 6).

Point 13: How do you measure the length of the weld pool?  Describe the routine to define it. And the glitters are the silicates from the filler material? (Figure 6 Line 188)

Response 13: Thanks very much for your careful review of our manuscript. We have redrawn Figure 6 (Line 197) to present the routine. The glitters were caused by the dynamic liquid level randomly generating a specular reflection between the laser and the camera. The silicates could not aggregate in large chunks at the front of the pool. As shown in section 3.3.

Point 14: How about the thesis of different weld pool temperature due to the different heat inputs between a), b) and c). the result in a different viscidity and different weld pool movement.

Response 14: Thanks very much for your careful review of our manuscript. The viscosity of molten metal in weld pool changes with different heat input. While the difference of the viscosity of the liquid metal caused by the small difference of heat input may not be the significant factor caused different movements of the weld pools.

Point 15: Formula (4) and (5) (Line 206) were wrong.

Response 15: Thanks very much for your careful review iof our manuscript. Under your guidance, we found this error and modified the formula (Formula (5) and (6) Line 215).

Point 16: Page 8 Line 236. The thermal accumulation of resistance could not be neglected.

Response 16: Thanks very much for your careful review of our manuscript. We are not phrasing properly here. Thank you for your preciseness. This sentence has been rewritten as “The energy accumulated in a very short time before the liquid bridge explosion is the main factor influencing the impact of electrical explosion. (Page 8 Line 245-247)

Point 17: please do a comparison of average power for a, b and c.

Response 17: Thanks very much for your careful review of our manuscript. Under your guidance, we compared the average power of three waveform control methods, as shown in Table 3 and Figure 20(a).

Point 18: where were the frequencies of weld pool from? Why the weld pool oscillation frequency of Cold Arc was not available?

Response 18: Thanks very much for your careful review of our manuscript. We have added some explanations in the article and hope to answer your questions(Line 296-299 Page 10). The contour diagram of the transform coefficient of the signal reflects the energy density distribution of the signal in the time-scale plane. The energy of the signal is mainly concentrated around the wavelet-ridge-cure in the time-scale plane, from which the instantaneous frequency of the signal can be determined. The scale vectors corresponding to the ridge-lines of small and medium waves in FIG. 13 can reflect the instantaneous frequency of weld pool oscillation, the formula describes the conversion relationship between the scale vectors and the instantaneous frequency of weld pool oscillation. However, no stable wavelet-ridge-cure can be found in Figure 13, so the frequency of Cold arc was not available.

Point 19: where is the metal oxide powder from? This is not the silicon oxide and manganese oxide?

Response 19: Thanks very much for your careful review on our manuscript. The white powder in the front and middle of the weld pool was the silicon oxide and manganese oxide particles, which are separated from the weld metal due to the strong turbulence in the weld pool in this part and pushed to the low-temperature area of the weld pool under the action of the pool flow.(Line 393-396, page 13)

Point 20: comparison to average welding power necessary, measure the areas of the penetration and compare these, the geometry is in relation to the welding power.

Response 20: Thanks very much for your careful review of our manuscript. Your reminding and guidance make us avoid a mistake, we made a comparison to average power and the areas of the penetration. As shown in Figure 20(a)(b). the main factor that determined weld geometry was welding heat input. The influence of weld pool behaviour on weld geometry and penetration area was not obvious.

Point 21: measure the particle size with, for example, the line cut method and compare it.

Response 21: Thanks very much for your careful review of our manuscript. The average grain sizes of AF grain were calculated using the intercept method (as per ASTM E112-10). And the grain size was shown in Figure 24.

Point 22: Conclusion 2,

Response 22: Thanks very much for your careful review of our manuscript. I'm sorry I didn't understand your comment, but I thought about where you underlined it. At the same welding speed, the main factor causing the decrease of the natural oscillation frequency of the molten pool is the increase in the volume of the weld pool. We modified the conclusion 2.

Point 23: Conclusion 4,

Response 23: After you suggested that we should consider the welding heat input and weld forming comprehensively, we recognized the mistake and revised conclusion 4. Compared with the influence of weld heat input on the size of the weld pool, the effect of weld pool oscillation is not obvious.

We hope you will be satisfied with the revisions for the resubmitted manuscript. If you have any queries and suggestions, please do not hesitate to contact me.

Many thanks for your time and consideration.

Best regards

Tao Chen

[email protected]

Reviewer 2 Report

The subject is worthy and interesting, and it is one to which the authors may add significant contributions, but the paper needs minor changes.

The main concerns and comments are given below:

In chapter " 2.2. Materials and Welding parameters", the authors should add the length of the welded plates and weld beads; show please the typical location and length of the high-speed imaging measurements performed on the bead. Please present and discuss the reliability and the validity of the measurements/results. 

In the same chapter the authors mentioned that "In order to acquire both partly and fully penetrated weld pools under the same welding parameters, 2mm and 4mm Q235 steel plate were selected as the base material.", please add and explain the joint type used in the study: butt-joint or bead-on-plate. The authors should discuss the admixture between the weld metal and base plate – add references.

Regarding Table 2 " Table 2 the effective heating power to the base material and weld pool outlines"; Add please a schematic view of the locations of the pool dimensional measurements including the accuracy and repeatability of the results. 

Please add to Figures 3 and 4 a general explanation as appears in Figures 5,6 etc. before (a), (b), etc.

"306 the fusion and partial weld pools of three waveforms, and table 2 shows", should be table 3.

For Tables 2 & 3 use please capital letters.

In " Figure 19. The geometry of the weld beads with different waveforms " add please explanations for (a) to (f) pictures.

The authors should add the information on the width and depth of the bead measurements locations " Figure 20. Effect of wire feed rate on geometry of fusion (partial penetration) in bead on plate deposition of Q235: (a) width; (b) depth." Elaborate on the accuracy of the dimensional measurements including the repeatability of the results.

"Figure 21. The microstructures of the weld beads with different waveforms (wire feed rate:3m/min)", the authors should significantly improve the pictures in Figure 21 and add high-magnification inserts to better show the PF and AF phases.

The authors mentioned that "481 Therefore, the volume fraction of acicular ferrite in figure 21 was measured, and the results were shown in Figure 22."  the authors should include detailed information regarding the measurement method of acicular ferrite volume and include the accuracy and the repeatability of the results. The authors should add "Grain Size Measurements of the Acicular Ferrite" in the weld metal microstructure.

The authors mentioned that " 529 increases, the content of acicular ferrite decreases while the grain size increases." The authors should show where and how they measured the weld metal "grain sizes".

I feel that the paper would benefit from the addition of references dealing with typical defects in the S-GMAW welds and by adding results on defects characterization (computed tomography or RT) present in the joints produced and used in this study.

I hope above comments help to improve a future version of the paper.

Author Response

Dear reviewer,

We deeply appreciate the time and effort you have spent in reviewing our manuscript. Based on your comments and suggestions, we have made modification on the manuscript and the revisions were highlighted in red in the revised manuscript. Here below are our descriptions of the revisions according to your comments.

Point 1: In chapter " 2.2. Materials and Welding parameters", the authors should add the length of the welded plates and weld beads; show please the typical location and length of the high-speed imaging measurements performed on the bead. Please present and discuss the reliability and validity of the measurements/results.

Response 1: Thanks very much for your careful review of our manuscript. The length of the welded plates and weld beads were added, the typical location and length of the high-speed imaging measurements performed on the bead were measured (Line 120-127, Page 4). The reliability and validity were discussed inline 111-114, Page 3

Point 2: In the same chapter the authors mentioned that "In order to acquire both partly and fully penetrated weld pools under the same welding parameters, 2mm and 4mm Q235 steel plate were selected as the base material.", please add and explain the joint type used in the study: butt-joint or bead-on-plate. The authors should discuss the admixture between the weld metal and base plate – add references.

Response 2: Thanks very much for your careful review of our manuscript. the joint type used in the study is bead-on-plate. And the chemical composition of the base material and filler wire are given in Table 1.

Point 3: Regarding Table 2 " Table 2 the effective heating power to the base material and weld pool outlines"; Add please a schematic view of the locations of the pool dimensional measurements including the accuracy and repeatability of the results.

Response 3: Thanks very much for your careful review of our manuscript. a schematic view of the locations of the pool dimensional measurements was added in Figure 6. the accuracy and repeatability of the results were added in Table 3.

Point 4: Please add to Figures 3 and 4 a general explanation as appears in Figures 5,6 etc. before (a), (b), etc.

Response 4: Thanks very much for your careful review of our manuscript. Under your guidance, we added a general explanation of Figures 3 and 4 as appears in Figures 5,6 etc. before (a), (b).

Point 5: "306 the fusion and partial weld pools of three waveforms, and table 2 shows", should be table 3.

Response 5: Thanks very much for your careful review of our manuscript. We found the error here and have corrected it, Thanks again for your careful review.

Point 6: For Tables 2 & 3 use please capital letters.

Response 6: Thanks very much for your careful review of our manuscript. We found the error here and have corrected it, Thanks again for your careful review.

Point 7: In " Figure 19. The geometry of the weld beads with different waveforms " add please explanations for (a) to (f) pictures.

Response 7: Thanks very much for your careful review of our manuscript. We found the error here and added explanations for (a) to (f) pictures., Thanks again for your careful review.

Point 8: The authors should add the information on the width and depth of the bead measurements locations " Figure 20. Effect of wire feed rate on the geometry of fusion (partial penetration) in the bead on plate deposition of Q235: (a) width; (b) depth." Elaborate on the accuracy of the dimensional measurements including the repeatability of the results.

Response 8: Thanks very much for your careful review of our manuscript. We remeasured the weld beads and redrew the pictures, the dimensional measurements and the repeatability of the results were presented in Figure 20.

Point 9: "Figure 21. The microstructures of the weld beads with different waveforms (wire feed rate:3m/min)", the authors should significantly improve the pictures in Figure 21 and add high-magnification inserts to better show the PF and AF phases.

Response 9: Thanks very much for your careful review of our manuscript. We added high-magnification figures in this paper (200 times), as shown in Figure 22.

Point 10: The authors mentioned that "481 Therefore, the volume fraction of acicular ferrite in figure 21 was measured, and the results were shown in Figure 22."  the authors should include detailed information regarding the measurement method of acicular ferrite volume and include the accuracy and the repeatability of the results. The authors should add "Grain Size Measurements of the Acicular Ferrite" in the weld metal microstructure.

Response 10: Thanks very much for your careful review of our manuscript. The area fraction of AF and PF phases in the metallographic picture were approximately equal to their respective volume fractions, and the results were shown in Figure 23. We remeasured the volume fraction of AF and PF phases and redrew the pictures, the dimensional measurements and the repeatability of the results were presented in Figure 23. The average grain sizes of AF grain were calculated using the intercept method (as per ASTM E112-10). Grain Size of the Acicular Ferrite was measured and added in Figure 24.

Point 11: The authors mentioned that " 529 increases, the content of acicular ferrite decreases while the grain size increases." The authors should show where and how they measured the weld metal "grain sizes".

Response 11: Thanks very much for your careful review of our manuscript. The average grain sizes of AF grain were calculated using the intercept method (as per ASTM E112-10). Grain Size of the Acicular Ferrite was measured and added in Figure 24.

We hope you will be satisfied with the revisions for the resubmitted manuscript. If you have any queries and suggestions, please do not hesitate to contact me.

Many thanks for your time and consideration.

Best regards

Tao Chen

[email protected]

Reviewer 3 Report

This paper provided a comprehensive study of the effects of three types of short-circuiting GWAM processes on weld pool behavior and microstructure. The three waveform behaviors were well explained in the introduction and the reasoning behind the work was explained. The experimental setup and methods to view and monitor the weld pool was carefully setup and done in an expert manner. The paper then provided a comprehensive and in-depth analysis of the weld pool behavior, showing a depth of knowledge by the authors.

The research provided; knowledge on how to view the weld pool behavior using high-speed photography, increased the communities understanding of the impact of three different waveform controls for S-GMAW on the weld pool behavior and also explained the significance of the results on the resultant weld penetration and microstructures. Therefore, this research added sound scientific knowledge to the welding community.

Overall, the paper was well laid out and easy to follow and minor grammar corrections are required.

Points to address

The red text in Figure 6 and 21 is difficult to read, especially Fig 21, as the test is invisible in the printed version. P10, L297, WFS was 2.1m/min, should it be 2.4? P10, Fig 11 needs a) to d) explained in the caption P 12 Fig 15, Caption should explain what the white dashed lines represented. P13 L398, “obvious differentiation’ should be explained in the text. P15 L468 FSP abbreviation needs to be explained, as well as how it relates to the PF phase, as FSP was not listed in Fig 21. P16, L483. Volume fraction method should be briefly explained.

Editorial changes

Formatting of Figures in the text made consistent, Ie. All with a capital 'F' P1 L41, this type of.. should be “The first method… P2 L45 …short-circuit period, dramatically reduces the short-circuit period. P2 L63 shows - showed P2 L74, However, no much research has been… P2 L77 replace ‘and hope’ with; weld pool, with the aim to reveal… P5 L146 remove ‘a’ in ‘single a droplet’ P6 L183 replace ‘the time of’ with the time when P7 L199 change sentence to, fluctuated the most, which was followed…. P7 L212 Figure 7 are the relationship curves that show… P7 Fig 7 caption …liquid bridge as a function of time. P7 L217 It has been pointed out… P7 L228 Figure 8 shows the relationship curves of the… P7 L229 The should be the P8 Fig 8 …liquid bridge as a function of time. P8 l256 .. amplitude of the weld pool was affected by the state of the weld pool; compared with… P9 L265 The relationship curves of the… P9 L271 …of the weld pool to change within a range. P10 Fig 11 Caption show describe a) to d). P10 L296 The relationship curve of the… P10 L304 Leaded should be led. (also P11 L335 & 347) P11 L315 Figure number not given P12 L361 weld pool was consisted… delete ‘was’. L362 weld pool is. P12 380-383. Paragraph awkwardly worded. The weld pool flow in partly the penetrated weld pool is explained with the assistance of Figure 16. When the weld pool is forced to flow downwards (by the external force), it is blocked by the base metal and is forced to flow to the back of the weld pool. At the back of the weld pool, the metal liquid flow will rebound off the solid metal interface and flow to the front of the pool. P13 L385 Leaded should be led to the formation of. P13 L388 ..the weld pool, because the bounced metal strength was negligible, so the metal oxides gathered into a… P13 L403 not pressed, so the… P13 L404 406 reword the sentence. For the fully penetrated weld pool, the slag at the end of the pool formed discrete scattered islands and did not gather into a large slag island, as was observed for the partial penetration case. This change was simply dependent on the… P14 L419 Delete it reveals in. Transverse sections are shown… P14 L430 leaded should be led to L431 pool boundary, which … P14 L432 Cold arc, so the area of the base metal fusion for the Cold arc.. P15 L462 For the same welding… P15 L465-6 weld bead, the greater the amount of PF formation. P15 L466-469 .. this phenomenon, as the full penetration weld was prepared with 2mm steel plate, which inhibited the heat transfer and reduced the cooling rate, it caused thicker PF and FSP (ferrite side plates ?) compared with the partly penetrated weld pool. P16 L484 .. significantly different, due to the change of the weld head gradient within the different waveforms.

Conclusions

L507 replace “, at the same time,” with ‘and’

L515 (2) delete ‘into’

Author Response

Dear reviewer,

We deeply appreciate the time and effort you have spent in reviewing our manuscript. Based on your comments and suggestions, we have modified the manuscript and the revisions were highlighted in red in the revised manuscript. Here below are our descriptions of the revisions according to your comments.

Point 1: The red text in Figure 6 and 21 is difficult to read, especially Fig 21, as the test is invisible in the printed version.

Response 1: Thanks very much for your careful review of our manuscript. We have redrawn Fig.6(line 198) and 21(line 482) to present a clear figure. The red text in these figures was changed in green and blue text.

Point 2: P10, L297, WFS was 2.1m/min, should it be 2.4?

Response 2: Thanks very much for your careful review of our manuscript. We found the error here and have corrected it, Thanks again for your careful review.

Point 3: P10, Fig 11 needs a) to d) explained in the caption.

Response 3: Thanks very much for your careful review of our manuscript. We added the explanation of a) to b) in the caption of Fig 11, Thanks again for your careful review.

Point 4: P 12 Fig 15, Caption should explain what the white dashed lines represented.

Response 4: Thanks very much for your careful review of our manuscript. “the slag islands were outlined by white dotted lines” was added in the caption of Figure 15(Line 387-88 )

Point 5: P13 L398, “obvious differentiation’ should be explained in the text.

Response 5: Thanks very much for your careful review of our manuscript. We added the explanation of “obvious differentiation”, This sentence “the full penetrated weld pools also have obvious low-temperature zone and high-temperature zone, but the metal oxides in the cold part of the weld pool did not gather and form huge slag islands, instead distributed at the back of the weld pool evenly.” was added in Line 420-423.

Point 6: P15 L468 FSP abbreviation needs to be explained, as well as how it relates to the PF phase, as FSP was not listed in Fig 21.

Response 6: Thanks very much for your careful review of our manuscript. FSP is the abbreviation of ferrite-side-plate. While in the microstructure of the weld bead in this study, the ratio of ferrite-side-plate was low, so we did not analyze it.

Point 7: P16, L483. Volume fraction method should be briefly explained.

Response 7: Thanks very much for your careful review of our manuscript. As shown in Line 510-511, The area fraction of AF and PF phases in the metallographic picture were approximately equal to their respective volume fractions.

We hope you will be satisfied with the revisions for the resubmitted manuscript. If you have any queries and suggestions, please do not hesitate to contact me.

Many thanks for your time and consideration.

Best regards

Tao Chen

[email protected]

Round 2

Reviewer 1 Report

Dear authors,

you improved the article a lot. I only marked hints reagrding the description of your figures and tables. In my opinion a Figure must be understandable only by looking to the figure and the undertitle. So please include relevant parameters in theundertitle of the figures and tables.

The rest you will see in my attached scaned version.

Author Response

Dear reviewer,

We deeply appreciate the time and effort you have spent in reviewing our manuscript. Based on your comments and suggestions, we have made modification on the manuscript and the revisions were highlighted in red in the revised manuscript. Here below are our descriptions of the revisions according to your comments.

Point 1: Image annotation needed to be changed (Figure 2 Line 87).

Response 1: Thanks very much for your careful review of our manuscript. We have redrawn Figure 2 (Line 87), wrong annotations have been modified.

Point 2: Page 5: Line 141: Grammatical mistakes

Response 2: Thanks very much for your careful review of our manuscript. We found the error here and corrected the grammar mistakes under your esteemed guidance.

Point 3: Page 5: Line 159: Lack of formula, tweld=tarc+tshort.

Response 3: Thanks very much for your careful review of our manuscript. We found the error here and added the missing formula under your esteemed guidance.

Point 4: Table 3 lacks information on wire feeding speed and plate thickness. (Page 6 Line186)

Response 4: Thanks very much for your careful review of our manuscript. We added the information of wire feed rate and base plate thickness in Table 3.

Point 5. Describe the method to find the border between solid and liquid.

Response 5: Thanks very much for your careful review of our manuscript. The area of the weld pool was measured by the Photoshop software, the border between the liquid and the solid was outlined manually, which could not be found by the software for tiny grayscale differences. (Page 7, Line 192-194)

Point 6. Image quality is too poor and no visible (Figure 5), The caption of the picture lacked the necessary information

Response 6: Thanks very much for your careful review of our manuscript. We have redrawn Figure 5 (Line 200) to present a clear annotation. We added the information of wire feed rate and base plate thickness in the caption of Figure 5.

Point 7. How do you calculate or measure V? Describe or show. (Page 7 Line220)

Response 7: Thanks very much for your careful review of our manuscript. The electrical explosion is caused by overheating of the metal at the neck of the bridge. The diameter of the liquid bridge changes gently in a small area near the neck constriction whose volume can be replaced by a cylinder whose diameter is equal with the diameter of the neck of the liquid bridge. (Page 7 Line 223-226).

Point 8 The caption of Figure 19 lacks relevant explanation (Page 15 Line 446).

Response 8: Thanks very much for your careful review of our manuscript. We redrew figure 19 and added the required annotations. (Page 15 Line 452)

We hope you will be satisfied with the revisions for the resubmitted manuscript. If you have any queries and suggestions, please do not hesitate to contact me.

Many thanks for your time and consideration.

Best regards

Tao Chen

[email protected]
